# Mechanism of the cadherin–catenin F-actin catch bond interaction

Amy Wang[1,2], Alexander R Dunn[1]*, William I Weis[2]*

[1]Department of Chemical Engineering, Stanford University, School of Engineering, Stanford, United States; [2]Departments of Structural Biology and Molecular & Cellular Physiology, School of Medicine, Stanford University, Stanford, United States

**Abstract** Mechanotransduction at cell–cell adhesions is crucial for the structural integrity, organization, and morphogenesis of epithelia. At cell–cell junctions, ternary E-cadherin/β-catenin/αE-catenin complexes sense and transmit mechanical load by binding to F-actin. The interaction with F-actin, described as a two-state catch bond, is weak in solution but is strengthened by applied force due to force-dependent transitions between weak and strong actin-binding states. Here, we provide direct evidence from optical trapping experiments that the catch bond property principally resides in the αE-catenin actin-binding domain (ABD). Consistent with our previously proposed model, the deletion of the first helix of the five-helix ABD bundle enables stable interactions with F-actin under minimal load that are well described by a single-state slip bond, even when αE-catenin is complexed with β-catenin and E-cadherin. Our data argue for a conserved catch bond mechanism for adhesion proteins with structurally similar ABDs. We also demonstrate that a stably bound ABD strengthens load-dependent binding interactions between a neighboring complex and F-actin, but the presence of the other αE-catenin domains weakens this effect. These results provide mechanistic insight to the cooperative binding of the cadherin–catenin complex to F-actin, which regulate dynamic cytoskeletal linkages in epithelial tissues.

*For correspondence:
alex.dunn@stanford.edu (ARD);
bill.weis@stanford.edu (WIW)

## Editor's evaluation

Single-molecule assays and kinetic modelling reported here validate and advance a structure-based model of the cadherin-catenin F-actin catch bond interaction, which is a fundamental cell-cell adhesive structure that can be both dynamic and force-activated. It is shown that the catch bond results from a force-dependent conformational change mechanism that may be conserved across other actin binding proteins.

## Introduction

The physical integrity and long-range organization of epithelial tissues are mediated in large part by dynamic linkages between intercellular adhesion complexes and the actomyosin cytoskeleton. Intercellular adhesions actively remodel in response to both external and cytoskeletally generated mechanical forces, both to reinforce tissues against forces that might otherwise threaten tissue integrity, and to drive cell–cell rearrangements that underlie embryonic morphogenesis and wound healing (*Charras and Yap, 2018*; *Ladoux and Mège, 2017*). Mechanotransduction at cell–cell adhesions likewise plays a central role in maintaining tissue homeostasis, and its dysregulation is associated with diseases such as metastatic cancer (*Ding et al., 2010*; *Vasioukhin, 2012*). Despite this physiological importance, the molecular mechanisms by which intercellular adhesions sense and respond to mechanical load remain incompletely understood.

Adherens junctions are essential intracellular adhesion sites in epithelial tissues. In these junctions, the extracellular domain of E-cadherins form contacts between neighboring cells, and their intracellular domains bind β-catenin. β-Catenin binds to αE-catenin, which binds directly to F-actin (*Desai et al., 2013*; *Meng and Takeichi, 2009*; *Rimm et al., 1995*; *Shapiro and Weis, 2009*; *Figure 1A*). The ternary E-cadherin/β-catenin/αE-catenin complex forms weak, transient interactions with F-actin in the absence of external load (*Drees et al., 2005*; *Miller et al., 2013*; *Yamada et al., 2005*). However, single-molecule force measurements revealed that mechanical force strengthens binding interactions between the ternary cadherin–catenin complex and F-actin (*Buckley et al., 2014*). This property, known as a catch bond, is thought to help reinforce intercellular adhesion under tension. Because the observed distribution of bond survival lifetimes between the cadherin–catenin complex and F-actin is biexponential, this interaction is best described by a two-state catch bond model defined by two distinct actin-bound states, weak and strong (*Buckley et al., 2014*). In this model, force enhances the transition from the weak to strong state, which results in longer binding lifetimes at higher load. Transitions between bound states are thought to arise from structural rearrangements in αE-catenin, which is allosterically modulated by binding partners and by mechanical load (*le Duc et al., 2010*; *Maki et al., 2016*; *Mei et al., 2020*; *Terekhova et al., 2019*; *Xu et al., 2020*; *Yonemura et al., 2010*). The catch bond interaction is also directional, such that force applied toward the pointed (−) end of the polar actin filament results in longer-lived bonds than when force is applied toward the barbed (+) end (*Arbore et al., 2022*; *Bax et al., 2022*).

αE-catenin consists of an N-terminal (N) β-catenin-binding domain, a middle (M) domain, and a flexible linker to a C-terminal actin-binding domain (ABD) (*Pokutta et al., 2014*; *Pokutta and Weis, 2000*; *Figure 1B*). Several lines of evidence suggest conformational changes within the αE-catenin ABD, a five-helix bundle (H1–H5) with a short N-terminal helix (H0), underlie catch bond formation (*Ishiyama et al., 2018*; *Ishiyama et al., 2013*; *Rangarajan and Izard, 2013*). Two recent structural studies (*Mei et al., 2020*; *Xu et al., 2020*) showed that whereas the structure of F-actin is essentially unchanged by complex formation, the ABD N-terminus through the last turn of H1 becomes disordered and helices H2–H5 repack, and the C-terminal extension (CTE, aa 844–906) that follows H5 becomes partially ordered and interacts with actin (*Figure 2A*). Consistent with the reported structures, we showed that removal of H0 and H1 produced 18× stronger binding of the ABD to F-actin in solution (*Xu et al., 2020*). Based on these structural and biochemical findings, we proposed that the observed four-helix, actin-bound ABD conformation represents the strong F-actin-bound state (*Figure 2B*).

Here, we test the structural model for catch bond formation with optical trapping measurements, which demonstrate that H0 and H1 of the αE-catenin ABD are required to confer directional catch bond behavior between the ternary cadherin–catenin complex and F-actin. Our findings are consistent with the structural model in which H0 and H1 reversibly undock from the remainder of the ABD to enable a transition between weak and strong actin-binding states. We further show that although the catch bond interaction is principally attributed to conformational changes in the ABD, the N and M domains of αE-catenin also regulate force sensitive binding.

## Results

### αE-catenin ABD and full-length monomer form a catch bond with F-actin

Previous investigations of the force-dependent binding of αE-catenin to F-actin have employed either the ternary E-cadherin/β-catenin/αE-catenin complex (*Bax et al., 2022*; *Buckley et al., 2014*), a binary β-catenin/αE-catenin complex, or αE-catenin alone (*Arbore et al., 2022*). The proposed molecular mechanism of the catch bond between αE-catenin and F-actin, however, is based upon structural data of the isolated and actin-bound αE-catenin ABD (*Xu et al., 2020*). To confirm that the catch bond behavior is truly associated with the ABD itself, we measured binding interactions between F-actin and the wild-type ABD (residues 666–906) under load with a constant-force assay in an optical trap, and compared our results to prior data (*Bax et al., 2022*) on the ternary complex obtained with the same instrument.

In the optical trap experiments, a taut actin filament is suspended between two trapped beads and positioned over αE-catenin ABD immobilized on microspheres that are attached to a coverslip

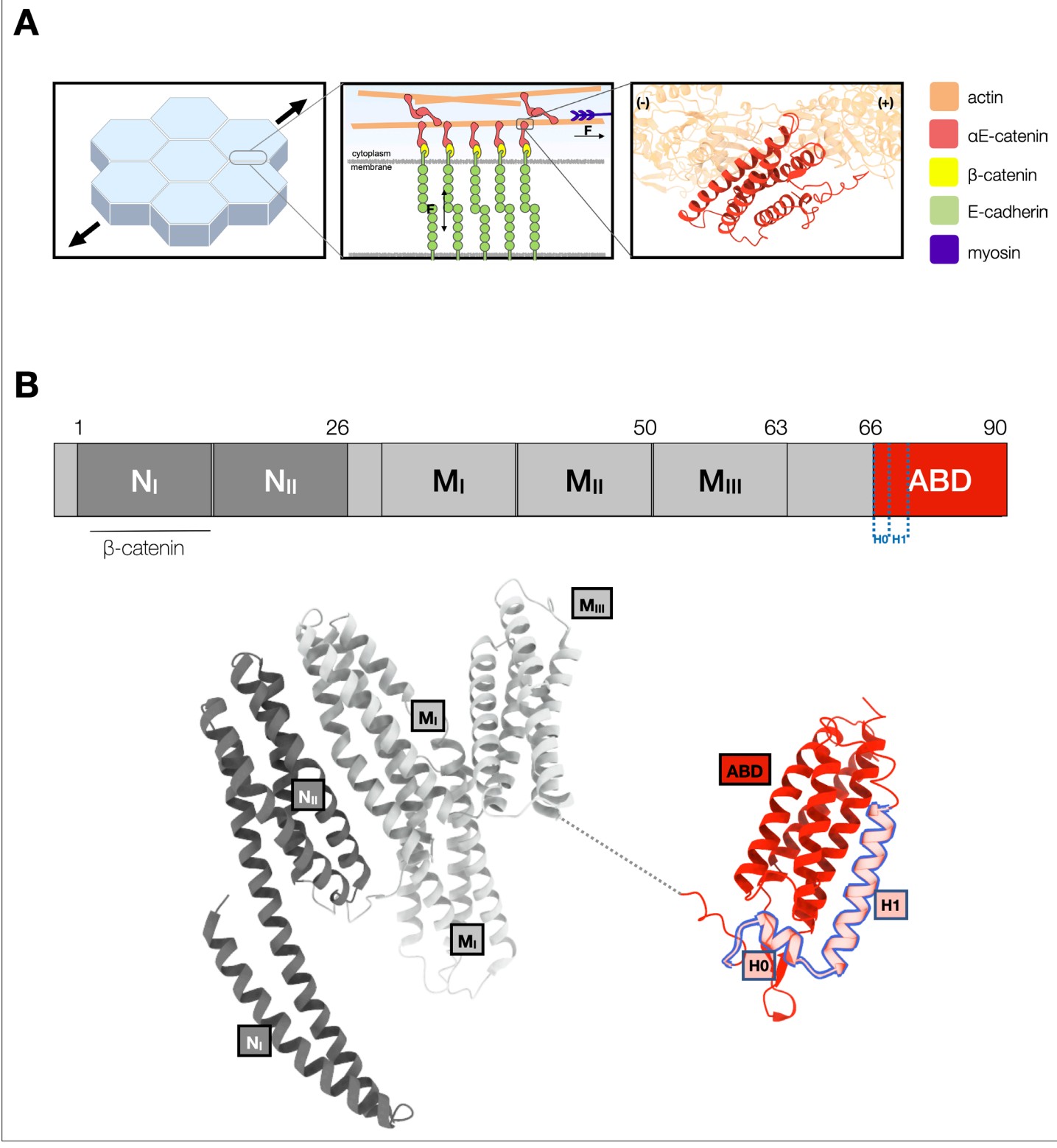

**Figure 1.** αE-catenin at adherens junctions. (**A**) Cell–cell adhesions in epithelia are reinforced under tension. Mechanotransduction at adherens junctions is mediated by both homophilic extracellular E-cadherin (green) interactions that establish adhesion between cells, and intracellular interactions of the cadherin–catenin complex with actin. Intracellularly, the cytosolic tail of E-cadherin binds to β-catenin (yellow) and αE-catenin (red) which forms a catch bond with F-actin. The structure of the αE-catenin actin-binding domain (ABD) complexed with F-actin is the basis for the catch bond mechanistic model. (**B**) Structure of full-length αE-catenin. αE-catenin (N and M domains: pdb 4igg) has a N-terminal domain that binds β-catenin, a middle (M) domain, and a flexible linker to the C-terminal ABD (pdb 6dv1). The ABD (red) is comprised of a five-helix bundle, preceded by a short N-terminal helix designated as H0, and a C-terminal extension (CTE). Helices H0 and H1 (residues 666–696) are outlined in blue.

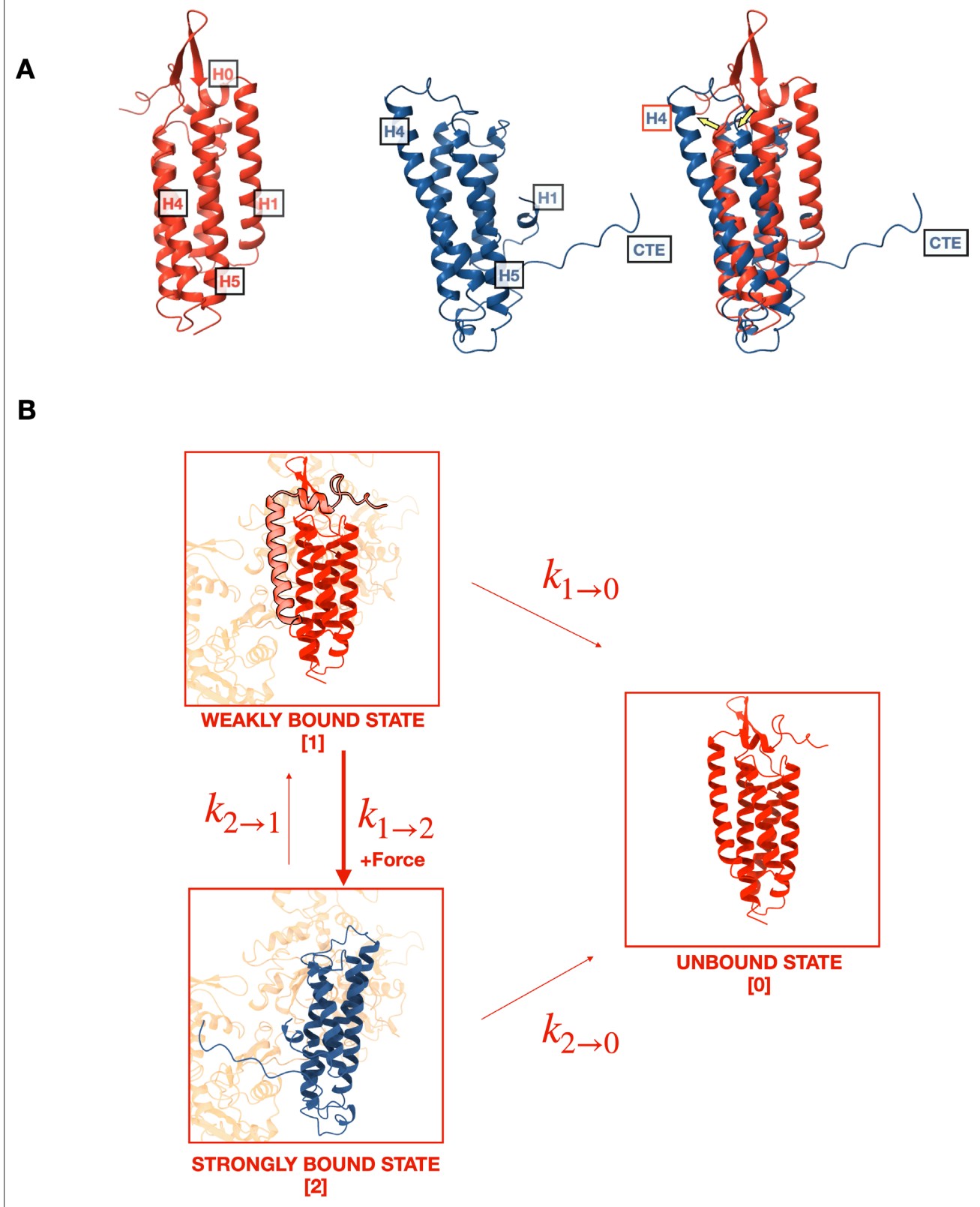

**Figure 2.** Mechanistic hypothesis of two-state catch bond from actin-binding domain (ABD) structures. (**A**) Comparison of isolated (red, pdb 6dv1) and actin-bound (blue, pdb 6upv) αE-catenin ABDs. In the actin-bound structure, H0 and H1 become disordered and H2–H5 rearrange (yellow arrows). The C-terminal extension (CTE), which is disordered in the isolated structure, forms an extended peptide and interacts with actin. (**B**) Two-state catch bond model. αE-catenin ABD interacts with F-actin in either the weak or strong conformational state, denoted as states 1 and 2, respectively. The unbound

*Figure 2 continued on next page*

*Figure 2 continued*

state is represented as state 0. The association of H0 and H1 with the four-helix bundle in the weak state (1) inhibits the ABD from rearranging into the strong state (2) conformation. The transitions between states are force dependent, and dissociation rates $k_{1 \to 0}$ and $k_{2 \to 0}$ increase exponentially with respect to applied load. Force also increases $k_{1 \to 2}$, the transition rate from states 1 to 2, but decreases $k_{2 \to 1}$. Tension applied to state 1 promotes the dissociation of H0 and H1 and the structural rearrangement of H2–H5 into state 2.

surface (*Figure 3A*). To exert load on the ABD, the stage is oscillated in a square wave parallel to the long axis of the filament. When a binding event occurs, the trapped beads are displaced from their equilibrium positions, resulting in a restoring force that can be measured with pN and ms resolutions. Stage motion pauses when bead displacement is detected, thereby applying a constant load to the bond between ABD molecules and F-actin (*Figure 3B*). As with wild-type ternary complex (*Bax et al., 2022*; *Buckley et al., 2014*), force on the beads commonly decreased in several discrete steps ('multi-step'), with each step corresponding to the release of a load-bearing molecule from the filament (*Figure 3—figure supplement 1*). The plateau of the final detachment step corresponds to the binding lifetime for the last remaining load-bearing molecule. Following full detachment, stage oscillation begins again, allowing us to collect multiple binding events from the same set of molecules.

The two-state catch bond model is described by the interconversion between a strongly bound state, a weakly bound state, and the unbound state (*Figure 2B*). The force-dependent interconversion rate between these states is given by the Bell–Evans model (*Bell, 1978*; *Evans and Ritchie, 1997*): $k_{i \to j}(\vec{F}) = k_{i \to j}^0 e^{F x_{i \to j}/k_b T}$, where $k_{i \to j}^0$ is the transition rate under no load, $\vec{F}$ is the force vector, and $\mathrm{x}$ is the distance between the initial state, $i$, with the transition state between $i$ and $j$ projected along $\vec{F}$. We used maximum likelihood estimation (MLE) to determine $k_{i \to j}^0$ and $x_{i \to j}$ parameters from observed binding lifetimes corresponding to the measured force. All binding lifetimes included in the analysis are derived from the final detachment plateau from multi-step data.

Our data revealed that the αE-catenin ABD forms a catch bond with F-actin, in which the lifetime of binding interactions increased with the application of mechanical force (*Figure 3C* and *Table 1*). The observation that the ABD forms a two-state catch bond to F-actin supports the structural model that the five- and four-helix conformation represents the weak and strong bound states, respectively. Previous modeling done by superimposing crystal structures of the isolated ABD on the actin-bound ABD structure showed few clashes with actin, suggesting that a similar five-helix structure may form a subset of interactions observed in the stably bound conformation (*Xu et al., 2020*). To quantify the possible differences in F-actin contacts between the proposed weak and strong state structures, we compared interactions between energy-minimized actin-bound ABD models and F-actin (*Figure 3—figure supplement 2*, *Supplementary file 1*). Energy minimization of the five-helix, ABD models resulted in approximately 0.5Å RMSD (root-mean-square deviation) compared to the undocked minimized structure, with the loop connecting H4 and H5 slightly repositioned to relieve minor clashes. The actin-bound four-helix ABD structure had a higher surface contact area than all three models of the docked ABD structures analyzed (*Supplementary file 1*), in part due to the CTE, which forms numerous interactions with actin in the bound structure (pdb 6UPV) but is otherwise disordered, as well as several residues in the extended H4 present in the actin-bound structure (*Mei et al., 2020*; *Xu et al., 2020*). Other residues in H4 and H5 observed to interact with actin in the actin-bound structure adopt similar positions in the five-helix bundle conformations. These observations are consistent with the proposal that a five-helix conformation similar to that of the isolated ABD can weakly interact with actin (*Xu et al., 2020*).

## H0 and H1 regulate the catch bond interaction between cadherin–catenin complexes and F-actin

To examine whether conformational changes in H0 and H1 of the ABD suffice to confer catch bond behavior to the interaction between the ternary cadherin–catenin complex and F-actin, we expressed and purified αE-cateninΔH1, in which residues corresponding to ABD H0 and most of helix H1 (residues 666–696) are deleted from the full-length protein (*Figure 4*). H2–H5 of the ABD is connected to αE-catenin N and M domains by the endogenous flexible linker, residues 633–665, consistent with the observation that H0 and H1 are disordered when the ABD is bound to F-actin (*Figure 2*).

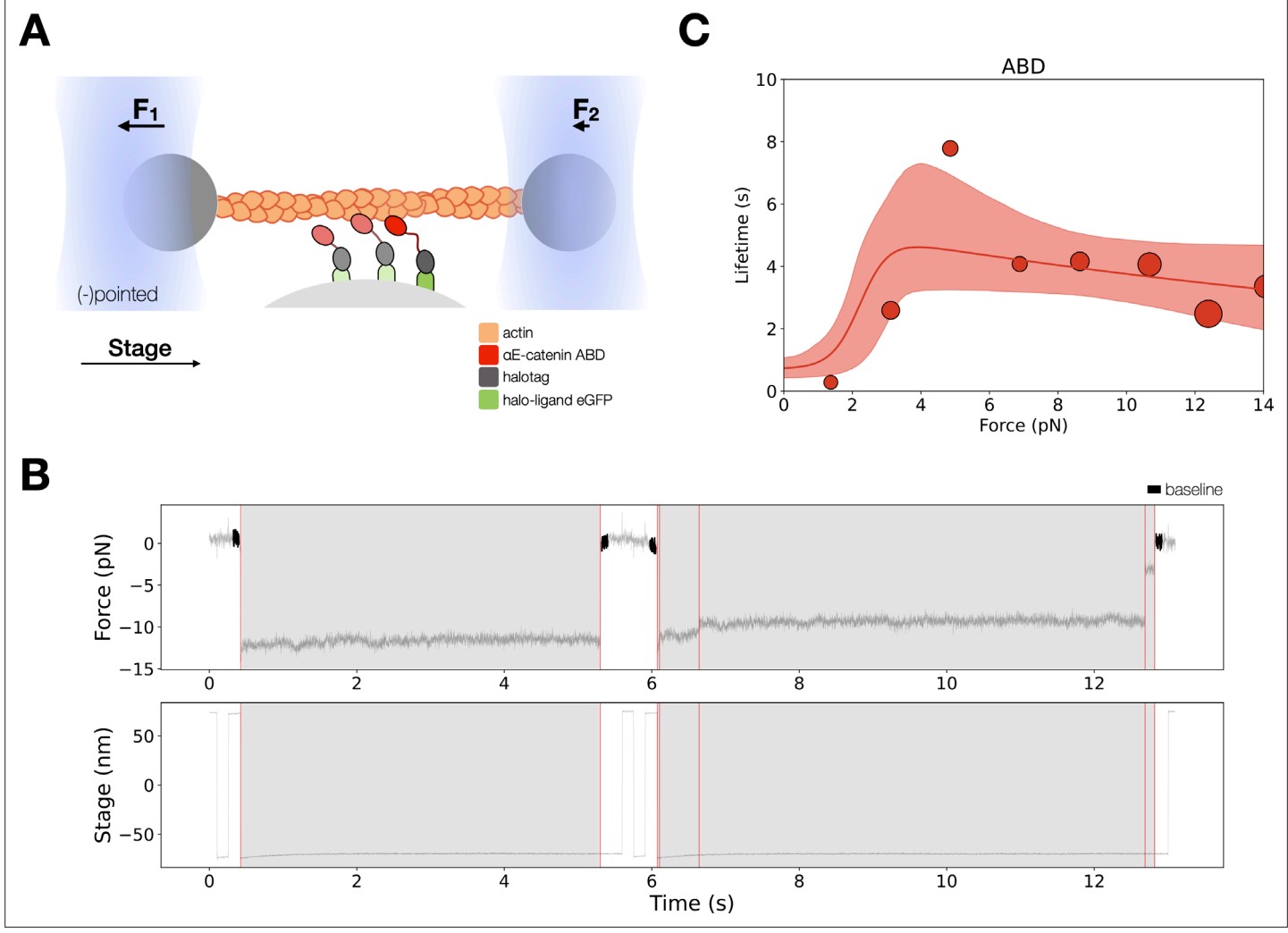

**Figure 3.** Force-dependent binding interactions between the αE-catenin actin-binding domain (ABD) and F-actin. (**A**) (Top) GFP-haloligand and fusion protein Halotag-ABD (red) complexes are immobilized on silica microspheres attached to a microscope coverslip. A taut actin filament is suspended between two optically trapped beads and held over the assembled complexes. The stage is translated parallel to the actin filament, and when at least one protein complex binds to F-actin, the trapped beads are pulled out of their equilibrium position. The restoring force of the optical trap (black arrows) applies tension on a bound complex while bystander complexes (pale) bind and unbind transiently. (**B**) A representative force versus time series for the constant-force assay. (Top) Plotted are the forces summed from both traps versus time, decimated from 40 to 4 kHz. We observe traces characterized either by rupture of a single bound molecule (left) or by sequential rupture of multiple bound molecules (right). Traces colored in black are regions used for force baseline determination, and vertical lines indicate step boundaries. (Bottom) If summed forces surpass a threshold, stage motion halts until detachment of the final bound molecule. (**C**) αE-catenin ABD forms a catch bond with F-actin (N = 900). Areas of all circles are proportional to the number of events measured in each equal-width bin. These data are represented here without depicting the direction of force applied relative to the polar actin filament.

The online version of this article includes the following source data, source code, and figure supplement(s) for figure 3:

**Source data 1.** αE-catenin actin-binding domain (ABD) optical trap constant-force assay multi-step force versus lifetimes.

**Figure supplement 1.** The distribution of the number of steps in constant-force assay measurements of ternary wild type (red, N = 1418), ternaryΔH1 (blue, N = 1604), and actin-binding domain (ABD) (gray, N = 1460).

**Figure supplement 1—source code 1.** Step number distribution analysis code for all events.

**Figure supplement 2.** Energy-minimized actin-binding domain (ABD) structures superimposed (red) or bound (blue) with actin.

**Figure supplement 3.** Force-dependent binding lifetimes of αE-catenin actin-binding domain (ABD) and monomer.

**Figure supplement 3—source data 1.** αE-catenin actin-binding domain (ABD) optical trap constant-force assay single-step force versus lifetimes.

**Figure supplement 3—source data 2.** αE-catenin monomer optical trap constant-force assay single-step force versus lifetimes.

**Figure supplement 3—source data 3.** αE-catenin monomer optical trap constant-force assay multi-step force versus lifetimes.

*Figure 3 continued on next page*

*Figure 3 continued*

**Figure supplement 3—source data 4.** Bootstrapped lifetime ratios for actin-binding domain (ABD) versus ternary complex multi-step data.

**Figure supplement 3—source data 5.** Bootstrapped lifetime ratios for αE-catenin monomer versus ternary complex multi-step data.

The structural hypothesis that undocking of H0 and H1 in the ABD is required to switch from the weak to strong-binding state predicts that αE-cateninΔH1 occupies a constitutively strong-binding state (*Xu et al., 2020*). To test this hypothesis directly, we performed optical trapping experiments to compare the force-dependent F-actin-binding lifetimes of the E-cadherin/β-catenin/αE-cateninΔH1 complex (ternaryΔH1) with those of the wild-type ternary complex. Here, αE-cateninΔH1 was assembled in a complex with full-length β-catenin and E-cadherin cytoplasmic domain tethered to the surface of the coverslip (*Figure 4A*). The same concentration of proteins used in the experiments with the wild-type complex caused all actin filaments in the flow cell chamber to absorb to the coverslip surface, so the ternaryΔH1 complex data were collected at a lower concentration. We note that the distribution in the number of steps observed in binding events is comparable between ternary wild-type and ternaryΔH1 complex datasets (*Figure 3—figure supplement 1*).

Two different optical trapping assays were employed to study the force-dependent binding of ternaryΔH1 complexes to F-actin. First, we used the constant-force assay described above to compare binding lifetimes for the ternary complexes assembled with wild-type αE-catenin and αE-cateninΔH1 (*Figure 4B*). Binding times for the wild-type ternary complex peak at ~6 pN, indicative of a catch bond (*Figure 5A*). In contrast, for αE-cateninΔH1, average binding times were highest at the lowest forces assayed and decreased with increasing load (*Figure 5B*). This latter observation is consistent with a simple Bell–Evans slip bond, in which load accelerates detachment from a single bound state.

Because the constant-force assay most frequently measured interactions between 4 and 8 pN, we employed a low-force assay in which the stage is moved sinusoidally at a low amplitude to measure binding under minimal load (*Huang et al., 2017*; *Figure 4C*). When a binding interaction occurs, the oscillation of the stage is transferred to the trapped beads, resulting in a detectable increase in its positional variance. The time-averaged force experienced by the optically trapped beads depends on the point at which binding occurs in the oscillation cycle, resulting in a distribution of forces between 0 and ~2.5 pN. Relative to the constant-force assay, measurements in the low-force assay may result in

**Table 1.** Kinetic parameters for the two-bound-state catch bond model for αE-catenin actin-binding domain (ABD) (top) and monomer (bottom).

State 0 is the unbound state, state 1 is the weak bound state, and state 2 is the strong bound state. The 95% confidence interval (CI) was determined for each parameter via empirical bootstrapping. The transition rate between state *i* and *j* under no load is indicated by $k^0_{i \to j}$, whereas the distance between the initial state, *i*, and the transition state between *i* and *j* is given by $x_{i \to j}$. A negative distance parameter indicates that the transition rate is decreased by force.

**ABD multi-step: two-state catch bond, nondirectional fit**

|  | 2 → 0 | 2 → 1 | 1 → 0 | 1 → 2 |
|---|---|---|---|---|
| $k^0_{i \to j}$ | 0.051 | 15.02 | 1.40 | 0.46 |
| CI (s⁻¹) | (0.02, 0.10) | (1.33, 1000) | (1.00, 2.46) | (0.18, 1.50) |
| $x_{i \to j}$ | 0.17 | 9.69 | 0.002 | 0.004 |
| CI (nm) | (0.01, 0.62) | (5.35, 21.42) | Fixed | (0.004, 0.145) |

**αE-catenin monomer multi-step: two-state catch bond, nondirectional fit**

|  | 2 → 0 | 2 → 1 | 1 → 0 | 1 → 2 |
|---|---|---|---|---|
| $k^0_{i \to j}$ | 0.046 | 0.72 | 4.97 | 0.87 |
| CI (s⁻¹) | (0.003, 0.39) | (0.40, 1000) | (3.41, 6.94) | (0.25, 2.75) |
| $x_{i \to j}$ | 0.64 | 0.66 | 0.002 | 0.76 |
| CI (nm) | (0.008, 2.29) | (0.08, 16.38) | Fixed | (0.004, 1.66) |

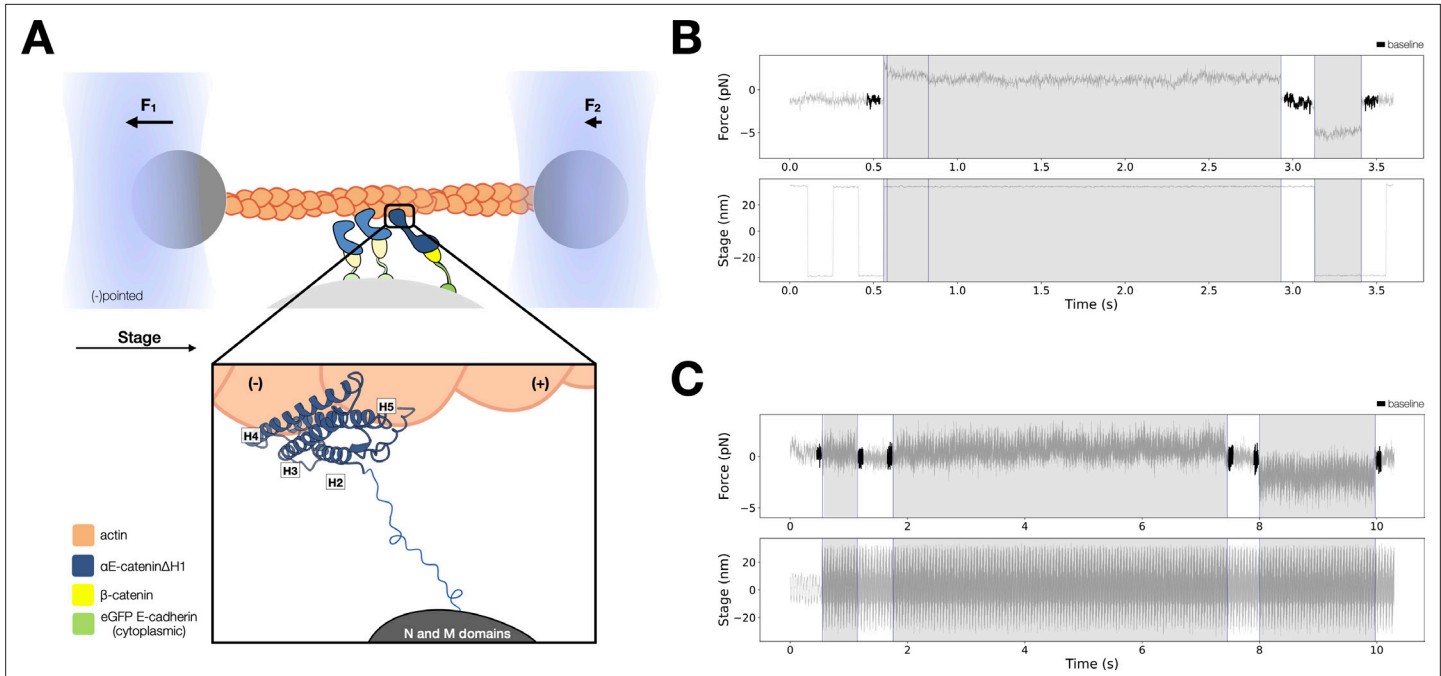

**Figure 4.** Force-dependent binding interactions between ternaryΔH1 complexes and F-actin. (**A**) Optical trap setup used in constant- and low-force assays. Ternary GFP-E-cadherin cytoplasmic tail (green), β-catenin (yellow), and αE-cateninΔH1 (blue) complexes are immobilized on silica microspheres attached to a microscope coverslip. (Inset) The actin-binding domain (ABD), which confers the catch bond interaction between cadherin–catenin complexes and F-actin, is attached to the M domain of αE-catenin by a flexible linker. The four-helix H2–H5 binds to actin directly in the purported strong state conformation. (**B**) Representative trace from the constant-force assay. (**C**) Low-force assay. (Top) A representative force versus time series (gray). Plotted are the forces summed from both traps versus time, decimated from 40 to 4 kHz. Binding lifetimes at low force were defined by the duration during which the positional variance of trapped beads exceeded the baseline variance of control experiments. Traces colored in black are regions used for force baseline determination. When a binding event occurs, stage motion is translated to the trapped beads. (Bottom) The stage oscillates in a high-frequency, low-amplitude sinusoidal waveform to enable binding event detection at low forces.

The online version of this article includes the following source data and figure supplement(s) for figure 4:

**Source data 1.** TernaryΔH1 optical trap constant-force assay multi-step data.

**Figure supplement 1.** E-cadherin/β-catenin/αE-cateninΔH1 low-force bond lifetimes.

**Figure supplement 1—source data 1.** TernaryΔH1 optical trap low-force assay data.

an overestimation of single-molecule-binding lifetimes due to the difficulty of resolving rupture events of multiple bound complexes. However, we found that the survival probability distribution from low-force assay measurements was not statistically different from that of constant-force measurements between 0 and 2.5 pN (*Figure 4—figure supplement 1*). Strikingly, the mean F-actin-binding lifetime for the ternaryΔH1 complex measured in the low-force assay is 2.4 s ($N$ = 145, 95% confidence interval [CI] = 1.9–3.0 s), 39 times longer than that of the wild-type complex (0.062 s; $N$ = 90, 95% CI = 0.036–0.095 s).

We used MLE to obtain $k^0_{U \to B}$ and $x_{U \to B}$ single-state Bell–Evans slip bond model parameters, where U and B are the unbound and bound states, from the ternaryΔH1 force-lifetime data (*Figure 5C*). Because the comparison of survival lifetime distributions indicates that the data from low-force assay measurements likely represent single-bond interactions, they were included in our analysis so that binding observations were sampled more evenly across the 0–8 pN range. Parameters $k^0_{U \to B}$ and $x_{U \to B}$ estimated for the ternaryΔH1 single-state slip bond model are consistent with strong-to-unbound parameters $k^0_{2 \to 0}$ and $x_{2 \to 0}$ estimated for the wild-type two-state catch bond model (*Table 2*). We also tested a model in which ternaryΔH1 complexes dissociate from two distinct bound states, B₁ and B₂, but goodness-of-fit assessed using both the Akaike information criterion (AIC) (*Akaike, 1974*) and Bayesian information criterion (BIC) (*Schwarz, 1978*) showed that the single-state slip bond model better represented the data (*Figure 5—figure supplement 1*, *Supplementary file 2*). Thus, our

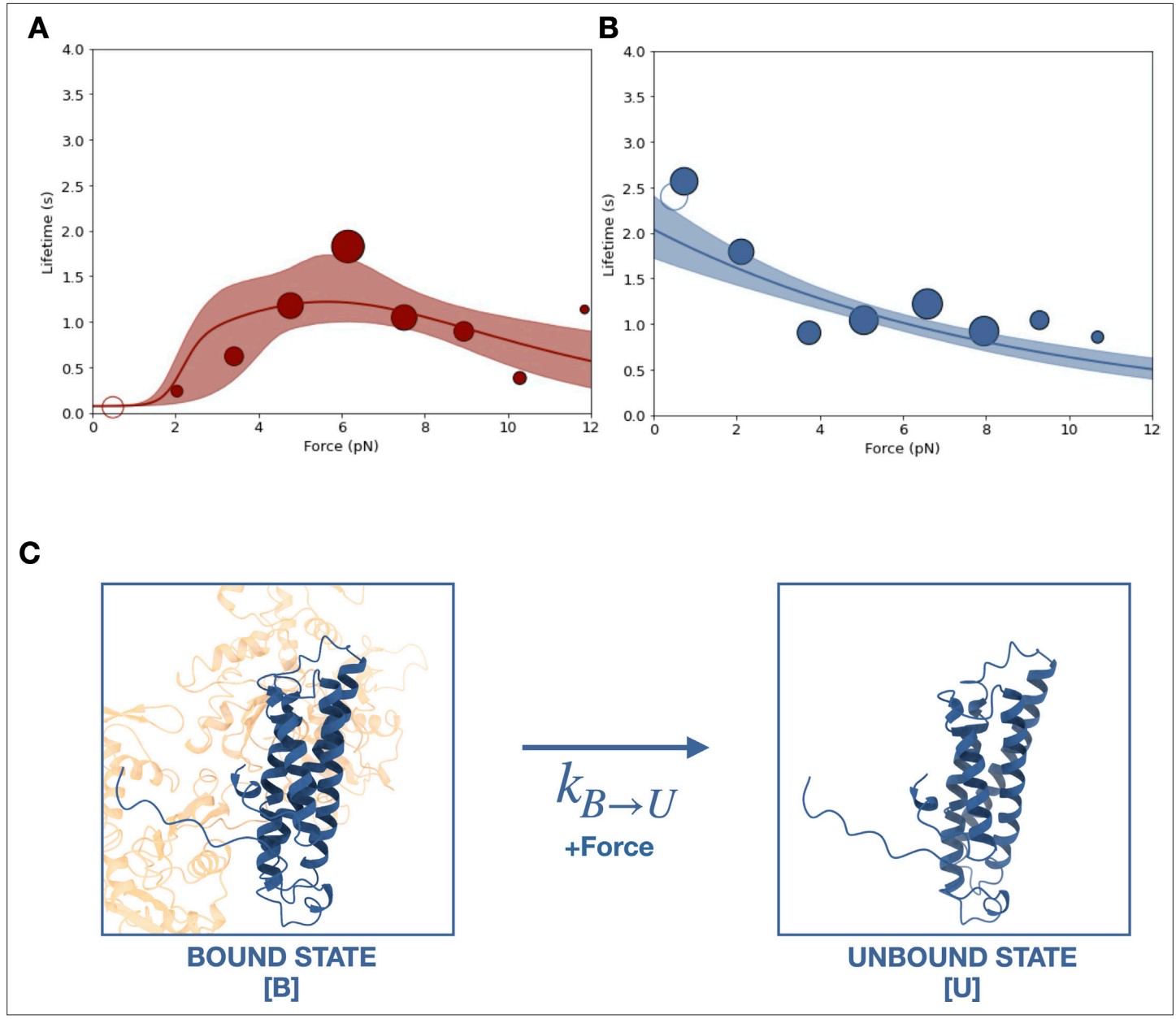

**Figure 5.** Force-dependent binding of cadherin/catenin complexes to F-actin. (**A**) Mean binding lifetimes (red filled circles) from constant-force assay measurements from previously reported (**Bax et al., 2022**) wild-type E-cadherin/β-catenin/αE-catenin complex data (N = 700). These data are represented here without depicting the direction of force applied relative to the polar actin filament, and fit to a nondirectional two-state catch bond (red curve). Unfilled circles represent the mean lifetime of events collected in the low-force assay (N = 90). Envelopes indicate 95% confidence intervals for the fit, obtained by empirical bootstrapping. Areas of all circles are proportional to the number of events measured in each equal-width bin. (**B**) Mean binding lifetimes (blue filled circles) from pooled low- (N = 145) and constant-force assay (N = 856) measurements for ternaryΔH1 complexes. These data were fit to a one-state slip bond model (blue curve). (**C**) The one-state slip bond model. The conformation of a bound actin-binding domain (ABD) missing H1, denoted state B, is comparable to the strong state of the two-state catch bond model. Molecules transition between bound (B) and unbound (U) states, where the dissociation rate, $k_{B \to U}$, increases exponentially with force.

The online version of this article includes the following source data, source code, and figure supplement(s) for figure 5:

**Source code 1.** Code for fitting and bootstrapping all slip bond models using maximum likelihood estimation.

**Source code 2.** Code for fitting a two-state catch bond model using maximum likelihood estimation.

**Source code 3.** Code for bootstrapping a two-state catch bond model using maximum likelihood estimation.

**Figure supplement 1.** Two-state slip bond model.

**Figure supplement 2.** Lifetime survival analysis for wild-type E-cadherin/β-catenin/αE-catenin and E-cadherin/β-catenin/αE-cateninΔH1.

*Figure 5 continued on next page*

*Figure 5 continued*

**Figure supplement 2—source data 1.** TernaryΔH1 optical trap low-force assay multi-step data.

**Figure supplement 2—source code 1.** Code for analyzing survival lifetimes across 2 pN force bins.

**Figure supplement 3.** Molecular basis of catch bond directionality.

**Figure supplement 3—source data 1.** Data file of binding events parsed by statistically inferred directionality.

**Figure supplement 3—source code 1.** Code for parsing and analyzing maximum directionality of ternary and ternaryΔH1 datasets.

analysis indicates that the deletion of H0 and H1 of the αE-catenin ABD eliminates the weak state conformation of the cadherin–catenin complex when bound to F-actin.

The observed binding lifetime distributions lend additional support to the model that the four-helix ABD bundle observed in the cryo-EM structures represents the strong F-actin-binding state. In the wild-type dataset, bond survival probabilities derived from constant-force measurements for each 2 pN force bin show biphasic distributions (*Figure 5—figure supplement 2*, *Supplementary file*

**Table 2.** Kinetic parameters describing force-dependent models for ternary wild type versus ternaryΔH1.

95% confidence intervals (CIs) for each parameter are obtained through empirical bootstrapping. Bound to unbound B → U single-state slip bond parameters for ternaryΔH1 correspond to the strong to unbound 2 → 0 two-state catch bond parameters for ternary wild type. Akaike information criterion (AIC) and Bayesian information criterion (BIC) indicated that the single-state slip bond model represented ternaryΔH1 data better than the two-state slip bond model.

**TernaryΔH1: one-state slip bond**

|  | B → U |
|---|---|
| $k^0_{i \to j}$ | 0.49 |
| CI ($s^{-1}$) | (0.41, 0.58) |
| $x_{i \to j}$ | 0.48 |
| CI (nm) | (0.36, 0.60) |
| AIC | 6.4 |
| BIC | 16.21 |

**Ternary wild type: two-state catch bond**

|  | 2 → 0 | 2 → 1 | 1 → 0 | 1 → 2 |
|---|---|---|---|---|
| $k^0_{i \to j}$ | 0.22 | 6.27 | 13.57 | 0.15 |
| CI ($s^{-1}$) | (0.15, 0.35) | (1.61, 369.66) | (13.27, 14.21) | (0.05, 0.39) |
| $x^{(-)}_{i \to j}$ | 0.55 | 3.46 | 0 | 4.72 |
| CI (nm) | (0.28, 0.78) | (1.30, 18.30) | Fixed | (3.92, 5.70) |
| $x^{(+)}_{i \to j}$ | 0.98 | 15 | 0 | 2.73 |
| CI (nm) | (0.74, 1.19) | Fixed | Fixed | (2.11, 3.40) |

**Ternary wild type: two-state catch bond, non-directional fit**

|  | 2 → 0 | 2 → 1 | 1 → 0 | 1 → 2 |
|---|---|---|---|---|
| $k^0_{i \to j}$ | 0.19 | 884.52 | 13.28 | 1.17 |
| CI ($s^{-1}$) | (0.05, 0.40) | (5.18, 1000.0) | (13.26, 14.37) | (0.08, 5.39) |
| $x_{i \to j}$ | 0.76 | 15.20 | 0 | 2.17 |
| CI (nm) | (0.37, 1.43) | (2.44, 17.53) | Fixed | (1.22, 4.55) |

3), consistent with the two-state catch bond model (*Barsegov and Thirumalai, 2005*; *Chakrabarti et al., 2017*; *Thomas et al., 2006*). In contrast, the distribution of survival probabilities for ternaryΔH1 complexes bound to F-actin appear monophasic below 6 pN, which likely indicates the presence of a single state in this force regime (*Figure 5—figure supplement 2*, *Supplementary file 3*). We noted no statistically distinguishable difference between ternaryΔH1 and wild type for survival likelihood distributions above 6 pN (*Supplementary file 4*), consistent with the idea that with wild-type complex predominantly occupies the strongly bound state at these forces.

The wild-type cadherin–catenin complex forms a directional catch bond with F-actin, where a higher binding frequency and larger extent of strong-state stabilization encoded by $x_{i \to j}$ is observed when force is applied toward the pointed (−) end of actin filament (*Bax et al., 2022*). In previous work, we hypothesized that the orientation of H1 could impart directionality: H1 would be pulled away from the H2–H5 bundle more readily when force was directed toward the (−) end of F-actin but would be relatively more aligned with purported weak state when force was directed toward the barbed (+) end (*Xu et al., 2020*; *Figure 5—figure supplement 3*). Likewise, reassociation of H1 with the rest of the ABD bundle would be less likely when subjected to (−) end directed force. To examine the *maximum* possible directional asymmetry present in the ternaryΔH1 complex dataset, we tabulated events for each actin filament as corresponding to either $F > 0$ or $F < 0$ in the reference frame of the optical trap, and assigned the inferred barbed (+) end to the group with the shorter mean lifetime. We then compared this upper bound on directionality for ternaryΔH1 with the assigned directionality of the wild-type ternary complex (*Bax et al., 2022*). For the ternary wild-type complex, the ratio of mean lifetimes between the implied (−) end versus (+) end is 3.78 (95% CI: 2.74–5.31), but 1.69 (95% CI: 1.38–2.09) for the ternaryΔH1 complex. Furthermore, the ratio of the number of binding events observed when force is oriented toward the implied (−) versus (+) end is 1.90 for the ternary wild-type complexes, but 1.13 for the ternaryΔH1 complex. These differences in lifetimes and numbers of observed events indicate that directionality is reduced in the ternaryΔH1 complex, consistent with the idea that the directional interaction between cadherin–catenin complexes and F-actin is largely attributable to the effect of force on the association/dissociation of H0 and H1.

## αE-catenin N and M domains allosterically regulate F-actin interactions

Binding of β-catenin to the αE-catenin N domain weakens the affinity of αE-catenin for F-actin (*Drees et al., 2005*; *Miller et al., 2013*; *Pokutta et al., 2014*; *Yamada et al., 2005*), and crosstalk between the ABD and the remainder of αE-catenin was detected in cysteine labeling experiments (*Terek-hova et al., 2019*). These observations indicate that allosteric coupling of the αE-catenin N, M, and ABD domains may affect actin-binding behavior. To assess whether such allostery affects the binding interactions with F-actin under force, we computed lifetime ratios (LRs) between the ABD, full-length αE-catenin monomer, and the ternary complex over a 4 pN sliding window force bin across 0–13 pN from our trap measurements. Strikingly, the ABD interaction with F-actin is fourfold longer than that of the ternary complex across all applied forces (mean LR = 4.28, 90% CI = 2.55–6.67), indicating that the N and M domains effectively destabilize actin binding (*Figure 3—figure supplement 3A–D*). The actin-binding lifetimes of monomeric αE-catenin (*Figure 3—figure supplement 3E–H*, *Table 1*) were comparable to those of the ternary complex (mean LR = 1.34, 90% CI = 0.77–2.20), demonstrating that β-catenin and the E-cadherin cytoplasmic domain do not impart the observed inhibitory contri-butions under load.

In the ternary complex experiments, multi-step events were still detected at the minimal concen-tration (5 nM) required to produce any binding events in the constant-force assay (*Buckley et al., 2014*), indicating that cadherin–catenin complexes preferentially bind actin when in the vicinity of other complexes. To simulate conditions in which stably bound untethered complexes are bound to nearby sites on actin, 100 nM ABD was added to the assay buffer; this produced many detect-able events with one apparent detachment ('single-step') even when the ternary complex was assem-bled at a concentration (1 nM) below the observable threshold (*Buckley et al., 2014*). Strikingly, the addition of soluble ABD also increased total binding lifetimes between actin and ternary complexes by approximately fourfold, congruent with our findings that the last-step binding lifetime between F-actin and ABD is fourfold longer than that of the ternary complex.

Given that cooperative interactions between neighboring ABDs enhance binding lifetimes (*Buckley et al., 2014*), we hypothesized that the presence of a stably bound neighbor might strengthen binding

interactions of a given complex and F-actin. Previous data indicate that one cadherin–catenin complex experiences most of the applied load while several other neighboring 'bystanders' transiently bind and unbind (**Bax et al., 2022**; **Figure 3A**). Thus, binding events that yield single-step observations likely reflect a load-bearing complex that is proximal to neighboring complexes that interact only transiently with actin. In contrast, the final load-bearing complex in a multi-step binding event is necessarily proximal to one or more complexes positioned such that they could form stable, force-bearing interactions with actin, implying an opportunity for cooperative binding interactions that could, in principle, influence binding lifetimes.

To determine whether neighboring protein complexes influenced F-actin binding, we compared the LRs for the last step of multi-step events versus single-step events across 0–13 pN with a 4 pN sliding window. Binding lifetimes from multi-step data are longer than single-step data for the ABD (mean LR = 3.54, 90% CI = 1.69–9.83) and αE-catenin monomer (mean LR = 3.04, 90% CI = 1.46–7.73) (**Figure 6A**, **Figure 6—figure supplement 1A**). These LR values are consistent with differences in two-state catch bond fits derived from single- versus multi-step binding events (**Figure 3—figure supplement 3**, **Supplementary file 5**). In contrast, differences in binding lifetimes between single- and multi-step data for the ternary complex (mean LR = 1.11, 90% CI = 0.78–1.67) and ternaryΔH1 complex (mean LR = 1.69, 90% CI = 1.34–2.15) were less pronounced (**Figure 6B**, **Figure 6—figure supplement 1**B). These observations suggest that force-induced proximal bystanders may allow the ABD to adopt conformations with more stable actin-binding characteristics (**Figure 6C**), but that interactions involving the N and M domains of αE-catenin, as well as β-catenin, inhibit cooperative binding interactions between neighboring complexes (**Figure 6D**; see Discussion).

## Discussion

We previously proposed a molecular mechanism (**Xu et al., 2020**) for the catch bond between actin and the cadherin catenin complex, wherein force promotes the dissociation of H0 and H1 from H2–H5 in the αE-catenin ABD, which allows the resulting four-helix H2–H5 bundle to rearrange and stably bind the actin filament with directional preference. Here, we provide direct experimental evidence from single-molecule optical trapping experiments that the catch bond interaction stems primarily from these conformational changes in the αE-catenin ABD (**Figure 3**). Additionally, our results show that in the absence of H0 and H1, the ternary cadherin–catenin complex, which otherwise transiently binds with F-actin in the absence of applied force, forms stable interactions (**Figure 5**). The ternaryΔH1 data can be thus described by a single-state slip bond with kinetic parameters consistent with those of the two-state catch bond model for the strong to unbound transition (**Figure 5**). In addition, ternaryΔH1-binding interactions appear to be less affected by the direction of force application with respect to the actin filament when compared to wild type. These data indicate an additional role for H0 and H1 in modulating the relative orientation of applied force and the reaction coordinate that characterizes the transition between the weak and strong states. More broadly, the rearrangement of the ABD from a five- to four-helix bundle necessarily alters the axis along which force is applied. This might be expected to alter the force-dependent actin dissociation of the strong versus weak state (**Le et al., 2021**).

A previous study proposed that removal of H0 (residues 670–673) enables stable F-actin binding by the αE-catenin ABD (**Ishiyama et al., 2018**). However, our biochemical and structural data demonstrated that removal of only H0 is unlikely to completely shift the conformational equilibrium of the ABD to the strong-binding state with the rearranged H2–H5 bundle (**Xu et al., 2020**). Nonetheless, epithelial monolayers formed by cells expressing αE-cateninΔH0 displayed more resistance to mechanical perturbation compared to wild type which, along with our in-solution measurements showing that deletion of H0 in the ABD moderately increased F-actin affinity, suggests that this may represent an intermediate to the strong state (**Xu et al., 2020**). Further studies will be required to determine the pathway by which H0 and H1 dissociate from the four-helix bundle and H2–H5 rearrange.

The molecular mechanism for the two-state catch bond described in this study may be conserved across several cell adhesion proteins. The five-helix bundle ABD of vinculin, an αE-catenin paralog that is a component of both integrin- and cadherin-based adhesions, also forms a directionally asymmetric two-state catch bond to F-actin (**Huang et al., 2017**). Although the vinculin ABD has no H0 and a shorter H1 compared to the αE-catenin ABD, cryoEM studies showed that, like αE-catenin, H1 is displaced from the five-helix bundle when bound to F-actin (**Kim et al., 2016**; **Mei et al., 2020**).

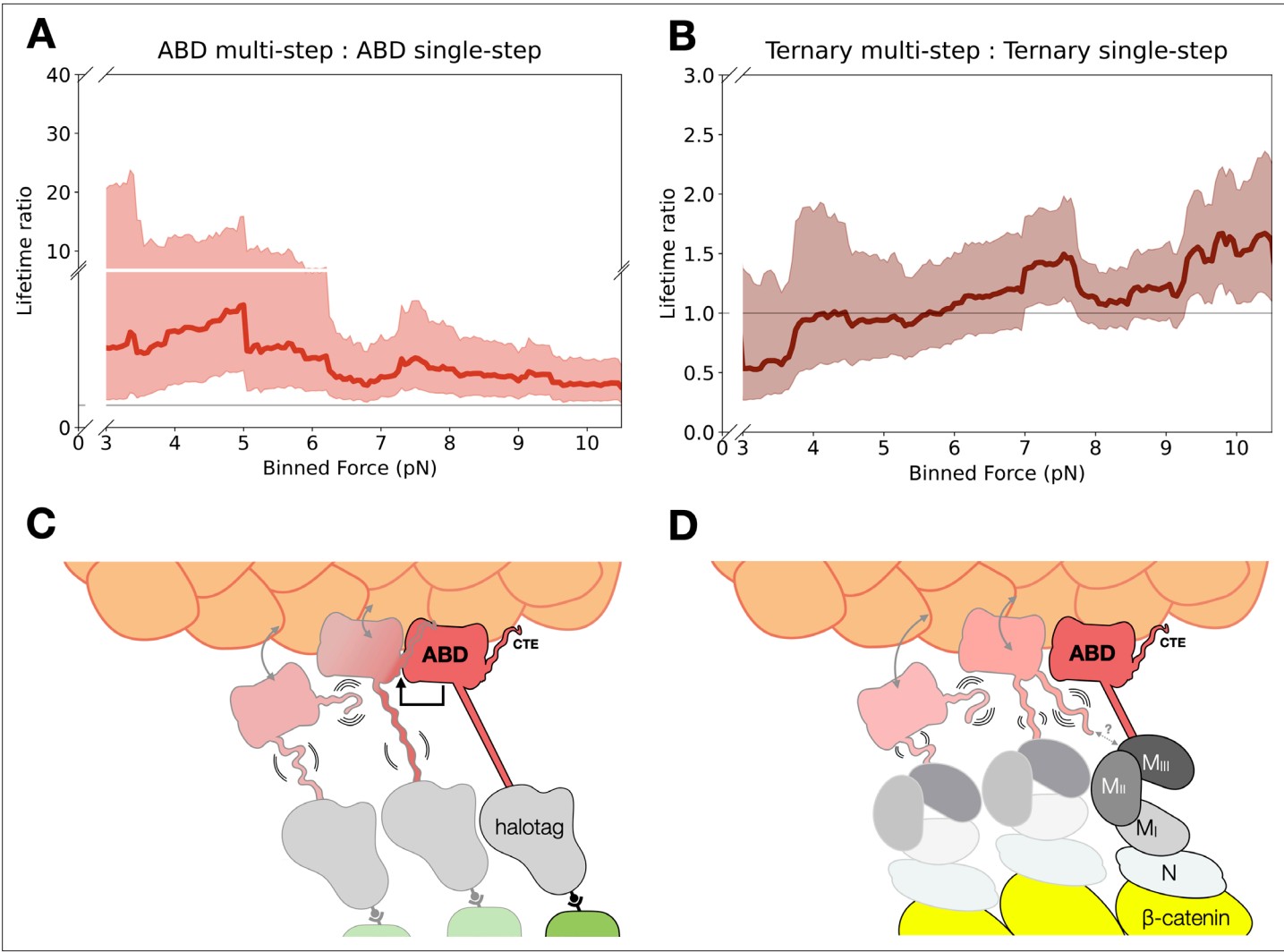

**Figure 6.** Model for cooperative binding under tension. (**A**) Computed lifetime ratios (LRs) with a 4 pN sliding window across 0–13 pN showing that lifetimes from actin-binding domain (ABD) multi-step events are longer than single-step events (mean LR = 3.54). Envelopes represent 90% confidence intervals (CIs), obtained via empirical bootstrapping mean (90% CI = 1.69–9.83). (**B**) Wild-type ternary lifetimes from multi- and single-step events have similar binding lifetimes (mean LR = 1.15, 90% CI = 0.68–1.78). (**C**) Upon stable binding with actin, a loaded ABD could enable stronger binding to actin by neighbors by allosteric coupling of involving contacts of the C-terminal extension (CTE) and the H2–H5 bundle. (**D**) The loaded ternary complex may interact with its neighbor differently than the ABD. Allosteric regulation of the ABD by the other αE-catenin domains, steric effects of the large N–M region, and/or differences in force propagation could prevent rearrangements in the ABD that would enhance its load-bearing capacity.

The online version of this article includes the following source data, source code, and figure supplement(s) for figure 6:

**Source code 1.** Code for analyzing and bootstrapping lifetime ratios.

**Source data 1.** Bootstrapped lifetime ratios for αE-catenin actin-binding domain (ABD) multi- versus single-step.

**Source data 2.** Bootstrapped lifetime ratios for ternary complex multi- versus single-step.

**Figure supplement 1.** Force-dependent cooperative binding for αE-catenin monomer and ternaryΔH1.

**Figure supplement 1—source data 1.** Bootstrapped lifetime ratios for αE-catenin monomer multi- versus single step.

**Figure supplement 1—source data 2.** Bootstrapped lifetime ratios for ternaryΔH1 multi- versus single step.

**Figure supplement 1—source data 3.** Ternary ΔH1 optical trap constant-force assay single-step force versus lifetimes.

Similarly, the C-terminal ABD of talin, which links integrins to the cytoskeleton in focal adhesions, forms a directionally asymmetric catch bond with F-actin (*Owen et al., 2022*). The talin ABD is a member of the THATCH family of actin-binding proteins, which consist of a five-helix bundle with an H1 that negatively regulates F-actin binding in solution-based assays (*Brett et al., 2006*; *Gingras et al., 2008*;

*Senetar et al., 2004*). Although the THATCH H1 packs against a different side of the H2–H5 bundle compared to the α-catenin/vinculin ABD (*Brett et al., 2006*), we speculate that the N-terminal helix release and the transition from a five- to four-helix bundle may be a mechanistically conserved feature that confers directional catch bonding to these ABDs. Although the biological function of directional catch bond formation remains speculative, its presence across multiple adhesion proteins is striking. One possibility is that the force-dependent directionality imparted by these interactions may serve to initiate or reinforce long-range order in the actin cytoskeleton, a possibility that is consistent with cell biological data and theoretical modeling (*Bax et al., 2022*; *Huang et al., 2017*; *Rahman et al., 2016*).

The αE-catenin ABD displays cooperative binding with actin, wherein a stably bound ABD enhances the binding stability of a proximal complex (*Figure 6*). This cooperativity may stem from rearrangements in H2–H5 and/or the CTE of the ABD. The CTE is largely disordered in the absence of actin, but in the actin-bound structure, V870 of the CTE packs against the C-terminal portion of the H4 extension in a neighboring ABD. Deletion of CTE residues 869–871, which removes the interaction of V870 with the neighboring ABD, resulted in no detectable actin binding in solution (*Xu et al., 2020*), suggesting that interactions between neighboring complexes are required to enter a nontransient actin-binding conformation. Cooperative binding was also observed in single-molecule optical trap assays (*Arbore et al., 2022*; *Buckley et al., 2014*). In particular, the addition of soluble ABD enhanced the binding lifetimes of surface-bound cadherin–catenin complexes approximately fourfold (*Buckley et al., 2014*). Thus, interactions between neighboring complexes on F-actin can be inferred to be important for stable F-actin binding either with or without applied load.

Importantly, it is probable that interactions between actin-bound neighbors are different for the ternary complex than for the isolated ABD. Binding to β-catenin dramatically weakens F-actin binding in solution assays (*Drees et al., 2005*; *Yamada et al., 2005*), indicating allosteric communication between the αE-catenin N and M domains and the ABD. Consistent with this interpretation, binding of E-cadherin/β-catenin to the N domain promotes conformational changes in H4 of the ABD as assessed by cysteine labeling (*Terekhova et al., 2019*). A full-length αE-catenin homodimer crystal structure (*Rangarajan and Izard, 2013*) showed that the CTE could pack either intermolecularly or intramolecularly with N and M domains, a result consistent with structural modeling based on small-angle X-ray scattering data (*Nicholl et al., 2018*). Previous biochemical data likewise indicate that the αE-catenin N and M domains inhibit cooperative actin binding by the ABD (*Hansen et al., 2013*). Thus, multiple lines of evidence suggest that contacts of the ABD and the N- and/or M-domains, perhaps involving the CTE, regulate the cooperativity of actin binding.

Our data likewise support a role for the αE-catenin N and M domains in modulating binding lifetimes under load. Binding lifetimes for the isolated ABD are approximately fourfold longer than for the ternary cadherin–catenin complex (*Figure 3—figure supplement 3*). Indirect evidence suggests this increase in stability may be coupled to cooperative interactions involving the CTE. Binding lifetimes for single-step ABD-binding interactions are roughly fourfold shorter than those that occur at the end of a multi-step unbinding sequence (*Figure 6A*), suggesting that neighbor–neighbor interactions can stabilize an actin-bound ABD (*Figure 6C*). In contrast, single- and multi-step lifetimes are comparable for the ternary complex (*Figure 6B*). A plausible explanation for this observation is that the N and/or M domains interfere with cooperative interactions between ABDs, perhaps via interactions with the CTE (*Figure 6D*). A role for the N and M domains in hindering stable binding to F-actin is consistent with optical trap results showing that an α–β-catenin heterodimer forms a transient slip bond with F-actin (<20 ms) in the absence of any bystanders, but that a longer-lived catch bond is recovered when multiple complexes are present (*Arbore et al., 2022*). Separately, we note that tension-dependent conformational changes in the N and M domains enable the recruitment of actin-binding partners such as vinculin (*Yao et al., 2014*; *Yonemura et al., 2010*), which can additionally mediate dynamic linkages and organization at cell junctions (*Bax et al., 2022*).

Whereas the directional catch bond mechanism for structurally similar actin-binding proteins is likely conserved, it is probable that the way in which force allosterically modulates actin interactions is variable. For example, although vinculin and αE-catenin are paralogs, structural and functional differences underlie their actin-binding characteristics. Helices H2–H5 of the vinculin ABD undergo similar structural rearrangements and share many contacts with actin as in αE-catenin, but their CTEs diverge in sequence and length and interact differently with actin (*Mei et al., 2020*). Additionally, when the autoinhibitory interactions formed between the N and C terminal regions of vinculin are disrupted,

actin-binding lifetimes for the full-length protein are enhanced twofold compared to the ABD alone (*Huang et al., 2017*), a trend opposite observed for αE-catenin (*Figure 3—figure supplement 3*). Diversity in actin binding and force transmission may be significant for maintaining intercellular adhesion or coordinating actin dynamics in tissues (*Clarke and Martin, 2021*; *Pollard, 2016*; *Svitkina, 2018*). Future analyses of how intermolecular and intramolecular interactions in other actin-binding proteins affect mechanotransduction will be required to understand how junctional tension is regulated at cell–cell contacts.

# Materials and methods

## Key resources table

| Reagent type (species) or resource | Designation | Source or reference | Identifiers | Additional information |
|---|---|---|---|---|
| Recombinant DNA reagent | pGEX-TEV | *Choi et al., 2012* | | Ampicillin resistance; expression in bacterial cultures; pGEX-KG plasmid (ATCC) with a new TEV protease site<br>Contact Weis lab for distribution |
| Strain, strain background (*Escherichia coli*) | BL21 (DE3) Codon-Plus RIL | Agilent | 230245 | Strain for expressing recombinant proteins |
| Chemical compound, drug | Halo-tag ligand Succinimidyl Ester O4 | | P6741 | Used for labeling GFP to attach to halotag constructs |
| Chemical compound, drug | Biotin-NHS | Millipore Sigma | P203118 | Labeling actin filaments |
| Chemical compound, drug | Rhodamine phallodin | Cytoskeleton | PHDR1 | Visualizing actin filaments |
| Chemical compound, drug | Trolox | Fischer Scientific | AC218940010 | |
| Chemical compound, drug | Bovine serum albumin, BSA | MCLAB | UBSA-100 | |
| Other | Streptavidin-coated polystyrene microspheres | Bangs Laboratories, Inc | CP01004 | |
| Software, algorithm | UCSF Chimera 1.14 | *Pettersen et al., 2004* | RRID:SCR_004097 | |
| Software algorithm | Python 3.9.1 | https://www.python.org/ | RRID:SCR_008394 | |
| Software algorithm | NumPy (v. 1.20.2) | https://numpy.org | RRID:SCR_008633 | Python library |
| Software algorithm | Pandas (v.1.3.1) | https://pandas.pydata.org | RRID:SCR_018214 | Python library |
| Software algorithm | SciPy (v.1.6.2) | https://scipy.org | RRID:SCR_008058 | Python library |
| Software algorithm | Matplotlib (v.3.5.1) | http://matplotlib.sourceforge.net | RRID:SCR_008624 | Python library |

## Protein expression and purification

Mouse GFP-E-cadherin cytoplasmic domain, and zebrafish β-catenin used in the optical trap assay were purified as described (*Bax et al., 2022*; *Buckley et al., 2014*; *Yamada et al., 2005*). αE-cateninΔH1 was constructed by inserting DNA encoding zebrafish αE-catenin with deleted H0 and H1 domains (aa 1–666 and 698–906) into the pPROEX HTb bacterial expression vector. αE-cateninΔH1 was expressed in *E. coli* BL21 (DE3) cells in 2 l LB media culture. Cells were grown at 37°C to an $OD_{600}$ of 0.8 before induced with 0.5 mM isopropyl-1-thio-β-D-galactopyranoside. After induction, cells were grown for 16 hr at 18°C, harvested by centrifugation, and resuspended in 20 mM Tris pH 8.0, 150 mM NaCl, 1 mM β-mercaptoethanol. Cell pellets were lysed with an Emulsiflex (Avastin) in the presence of protease inhibitor cocktail Mixture Set V (Calbiochem) and DNAse (Millipore Sigma). The lysate was clarified by centrifugation at 37,000 × *g* for 30 min, and incubated with 10 ml of TALON Superflow resin (GE Healthcare Life Sciences) for 1 hr on a rotator at 4°C. Resin was washed with 5 bed volumes of 20 mM Tris pH 8.0, 150 mM NaCl, 1 mM β-mercaptoethanol, 4 bed volumes of phosphate-buffered saline pH 8.0, 1 M NaCl, 0.005% Tween 20, and 3 bed volumes of 20 mM Tris pH 8.0, 150 mM NaCl, 1 mM β-mercaptoethanol, 5 mM imidazole. Protein was eluted from TALON resin in 20 ml of 20 mM

Tris pH 8.0, 150 mM NaCl, 1 mM β-mercaptoethanol, 150 mM imidazole. The eluate was passed through a 0.22-μm PES syringe filter and diluted with 20 mM Tris pH 8.0, 1 mM DTT (Dithiothreitol) to a final volume of 70 ml. Filtered eluate was further purified on an anion exchange column (MonoQ 10/100, GE Healthcare) in 20 mM Tris pH 8.0, 1 mM DTT buffer with a 0–1 M NaCl gradient, followed by size exclusion chromatography (Superdex S200, GE Healthcare) in 20 mM Tris pH 8.0, 150 mM NaCl, 1 mM DTT. Proteins were stored at −80°C and never underwent more than one freeze/thaw cycle.

αE-catenin ABD and αE-catenin monomer measurements with F-actin in the optical trap assay were performed with a Halotag-ABD construct. The halotag-ABD and halotag-αE-catenin monomer construct was constructed by inserting DNA encoding HaloTag, an 18 residue linker (SGGGGSGG-GGSGGGGSGG) and either the ABD domain (aa 666–906) or full sequence (aa 2–906) of zebrafish αE-catenin into the pPROEX HTb bacterial expression vector. Halotag-ABD was expressed and purified as described for αE-cateninΔH1. eGFP was purified and labeled with halotag Succinimidyl Ester (O4) ligand as previously described (*Huang et al., 2017*) (Promega).

## Preparation of fluorescent biotinylated F-actin

Actin was purified from rabbit skeletal muscle, stored and biotinylated using biotin-NHS (Sigma) as previously described. The biotinylated actin was flash frozen at 24 μM in G-buffer (5 mM Tris pH 8.0, 0.2 mM $CaCl_2$, and 0.2 mM ATP) with 1 mM DTT and stored in −80°C. G-actin was thawed on ice for 30 min and centrifuged in a Beckman TLA100 rotor at 60 k rpm for 10 min in 4°C to remove aggregates. Polymerization of G-actin was induced upon addition of 10× F-buffer (100 mM pH 7.5 Tris, 500 mM KCl, 20 mM $MgCl_2$, 10 mM ATP, 10 mM DTT) and incubated for 1 hr at room temperature while on a rotator. F-actin was diluted to 3.5 μM with F-buffer (20 mM Tris 8, 50 mM KCl, 2 mM $MgCl_2$, 0.2 mM $CaCl_2$, 1 mM DTT, 1 mM ATP). Lyophilized rhodamine phalloidin (Cytoskeleton) was resuspended with methanol (ACS Spectrophotometric Grade, ≥99.9%, Honeywell) and added to F-actin in equimolar amounts. Fluorescent biotinylated F-actin was kept on ice at 4°C for at least 1 day before use in experiments to allow rhodamine phalloidin to incorporate into filaments and used in optical trapping experiments within 10 days. Aliquots from the same batch of biotinylated actin were used in all single-molecule experiments.

## Flow cell preparation

Nitrocellulose-coated coverslips with attached 1.5-μm silica microspheres (Bangs Laboratories) and flow cell chambers were prepared as described previously for all optical trap experiments (*Huang et al., 2017*; *Bax et al., 2022*). All injection volumes were 10 μl. The flow cell channel was injected with F-buffer. GFP-E-cadherin was injected and allowed to nonspecifically absorb onto the coverslip and silica microsphere surfaces before being washed out with F-buffer following a 2-min incubation. For surface passivation, 5% (wt/vol) pluronic F-127 (Sigma, P2443) in F-buffer was injected and incubated for 5 min, twice. F-buffer was injected into the channel to wash out excess pluronic, twice. β-Catenin was injected into the channel and incubated for 2 min. Excess protein not bound to immobilized E-cadherin was washed out twice with F-buffer. αE-catenin was subsequently injected and incubated for 2 min, where excess protein was washed out twice with F-buffer. 1 mg/ml ultrapure bovine serum albumin (BSA; MCLAB, UBSA-100) in F-buffer was injected and incubated for 2 min, twice. The channel was finally filled with a trapping solution of 1 mg/ml BSA, 2 mM protocatechuic acid (Sigma-Aldrich), 50 nM protocatechuate-3,4-dioxygenase (Sigma-Aldrich), 1 μM Trolox (Sigma-Aldrich), 20 μM phallodin, 1 μm streptatvidin-coated polystyrene beads (Bangs Laboratories), and 0.2 nM fluorescently labeled biotinylated actin filaments. After the final solution was injected into the flow cell, the open ends of the channel were sealed with vacuum grease (Dow Corning).

While surface functionalization of wild-type cadherin–catenin complexes was prepared by subsequent injection of 50 μM GFP-E-cadherin, 100 nM β-catenin, and 75 nM wild-type αE-catenin, all F-actin filaments present in the flow cell were specifically adsorbed to the coverslip surface when flow cells were prepared with αE-cateninΔH1. Thus, to functionalize surfaces with E-cadherin/β-catenin/αE-cateninΔH1 complexes, 20 μM GFP-E-cadherin, 100 nM β-catenin, and 75 nM αE-cateninΔH1 were subsequently injected through the flow cell chamber.

Flow cell chambers for αE-catenin ABD experiments were prepared by described above, but with subsequent injection of 10 µM haloligand-eGFP and 2 µM halotag-ABD. Measurements αE-catenin monomer were similarly carried out but with 1 µM halotag-αE-catenin.

## Optical trap instrument

The optical trap instrument used was described previously (*Bax et al., 2022*; *Buckley et al., 2014*; *Huang et al., 2017*; *Owen et al., 2022*). Bead displacement was calibrated within the linear region of the quadrant photodiode voltage response for position detection. A stiffness calibration for each trap was performed using power spectral analysis according to previously established methods (*Berg-Sørensen and Flyvbjerg, 2004*; *Hansen et al., 2006*). The trap was operated at a stiffness of 0.1–0.15 pN/nm.

## Constant-load optical trap assay

The dual-beam optical trap assay was carried out as described (*Bax et al., 2022*; *Buckley et al., 2014*; *Huang et al., 2017*; *Owen et al., 2022*). Two optically trapped streptavidin-coated polystyrene beads (1 µm, Bangs Laboratories) were moved apart until 1–3 pN of tension was applied to a tethered biotinylated F-actin filament. The F-actin filament was centered near a surface immobilized silica microsphere (1.5 µm, Bangs Laboratories) functionalized with ternaryΔH1 complexes. The instrument stage was then oscillated in a trapezoidal waveform with 20–75 nm amplitude, 10 nm/ms rise/fall rate, and a 150-ms pause to check for displacement of either trapped bead from binding of cadherin–catenin complexes to F-actin. If a binding event was detected, the stage paused oscillation until trap signal returned to baseline values when all bound complexes released from the filament. Bead position data were collected from each trap at a sampling rate of 40 kHz, and down sampled to 1 kHz for force-lifetime analysis. Binding events that did not survive the 5-ms loading phase or resulted in dumbbell slackening of over 1.5 pN were excluded from further analysis. Control experiments in which surface-functionalized silica microspheres were functionalized with E-cadherin/β-catenin resulted in no binding activity.

Binding events were annotated with custom software (Python) by edge detection analysis, where data traces were convolved with the second order derivative of the Gaussian kernel and change points were identified at zero-crossings (*Haralick, 1987*). All events and steps were verified or reannotated manually.

Force associated with each binding event was calculated as previously described (*Bax et al., 2022*; *Huang et al., 2017*).

## Low-force optical trap assay

Assembly of suspended F-actin filaments, and binding activity determination of cadherin–catenin complexes to F-actin was performed as for a constant-load optical trap assay. After verifying binding activity of ternaryΔH1 complexes functionalized on silica microspheres, the instrument stage was oscillated in a sinusoidal waveform with 20–30 nm amplitude and 150 Hz frequency without force-feedback control. Data collected where the positioning of microspheres relative to F-actin resulted in no binding activity was used to establish a baseline in event detection analysis.

Signal from each trap collected at 40 kHz was down sampled to 1 kHz. A power spectrum of the sum of bead positions from both traps was computed using a Fourier transform with a moving window of 256 points. The cumulative power from frequencies higher than 300 Hz was calculated at each point and used to determine low-force binding events, as described previously (*Huang et al., 2017*). Deviations of the summed high-frequency power above 180% of the mean were labeled as a binding event. Binding was often accompanied by a change in the mean position of the trapped beads, resulting in a net force ranging from 0.2 to 2.5 pN exerted on the F-actin filament. Control experiments where silica microspheres were functionalized with E-cadherin/β-catenin resulted in no binding activity.

## Model fitting and CIs

All binding lifetime data fits were derived from the last step of multi- or single-step data from the constant-force assay. Best-fit parameters for slip bond models were determined by MLE on the individual force-lifetime ternaryΔH1 measurements, pooled from constant- ($N$ = 856) and low-force ($N$ = 145) observations. All objective function minimizations were performed using the SciPy optimize

minimization routine with a L-BFGS-B algorithm. The likelihood function for a slip bond model was: $\mathcal{L}(\theta|\mathrm{F},\tau) = \mathrm{k_{B\rightarrow U}(F)e^{-k_{B\rightarrow U}(F)\tau}}$, where $\tau$ and $F$ represent the bond lifetimes with respect to force measured in single-molecule experiments and $\theta$ are best-fit parameters. The likelihood function for a two-state slip bond was: $\mathcal{L}(\theta|\mathrm{F},\tau) = \mathrm{P_1 k_{B_1\rightarrow U}e^{-k_{B_1\rightarrow U}(F)\tau}} + (1-\mathrm{P_1})\mathrm{k_{B_2\rightarrow U}(F)e^{-k_{B_2\rightarrow U}(F)\tau}}$ where $\mathrm{B_1}$ and $\mathrm{B_2}$ represent the two distinct bound states and $\mathrm{P_1}$ represents the probability of observing a binding event in state 1. All $k_{i\rightarrow j}(\mathrm{F})$ parameters were described by the Bell model, $k_{i\rightarrow j}(\mathrm{F}) = k_{i\rightarrow j}^0 e^{\mathrm{F}x_{i\rightarrow j}/k_b\tau}$. CIs were determined by empirical bootstrapping, where each of the 10,000 synthetic datasets were constructed by drawing $N$ = 1001 force versus lifetime observations from the ternaryΔH1 dataset with replacement and fit to a model by MLE.

MLE objective function minimizations were similarly performed for the two-state catch bond model, and as described previously (*Bax et al., 2022*; *Huang et al., 2017*). With a Matlab implementation of the genetic algorithm,100 epochs were used to find a global minima for each dataset and the eight-parameter fits were constrained such that the mean lifetime at zero force was less than or equal to 100 s.

95% CIs on the parameters were determined by identifying parameter values in the 2.5th and 97.5th percentile. CI bounds on the model were determined as 95% CIs of binding lifetimes,$< \tau >$ , at each force predicted by the fits from the 10,000 synthetic datasets, evaluated as $< \tau > = \int_0^\infty \mathrm{t} \times \mathcal{L}(\theta|\mathrm{F},\tau)\mathrm{dt}$. For bootstrapped fits of a two-state catch bond model, 20 epochs of the genetic algorithm were used to find the minima of each synthetic dataset.

## Structure minimization and analysis

Maestro (Schrödinger) was used to perform energy minimization (OPLS 2005 force field) on isolated ABD structures and actin-docked structures. C $\alpha$ RMSD to 6UPV ABD was calculated for aa 711–842. Surface area of actin-binding interfaces was calculated in PyMOL, and RMSD and binding interaction analysis were carried out in ChimeraX for all energy-minimized structures. Structural figures were prepared with UCSF ChimeraX version 1.3.

## Materials availability

Requests for resources and reagents should be directed to the corresponding author, William I. Weis (bill.weis@stanford.edu). All reagents generated in this study are available without restriction.

## Acknowledgements

We thank Dr. Nicholas A Bax for his contributions in the early stages of this project and Dr. Chao Liu for assistance with instrument alignment. Research reported in this publication was supported by a Howard Hughes Medical Institute Faculty Scholar Award (ARD), National Institutes of Health grant R01GM114462 to WIW and ARD, R35GM130332 to ARD and R35GM131747 to WIW. AW is supported by the National Science Foundation Graduate Fellowship, the Stanford Graduate Fellowship and NIH training grant T32GM120007. The SSRL Structural Molecular Biology Program is supported by the Department of Energy Office of Biological and Environmental Research and by the National Institutes of Health, NIGMS Grant P30GM133894. The contents of this publication are solely the responsibility of the authors and do not necessarily represent the official views of NIGMS or NIH.

## Additional information

### Competing interests
William I Weis: Reviewing editor, eLife. The other authors declare that no competing interests exist.

### Funding

| Funder | Grant reference number | Author |
| --- | --- | --- |
| National Institutes of Health | R01GM114462 | Alexander R Dunn |

| Funder | Grant reference number | Author |
|---|---|---|
| National Institutes of Health | R35GM130332 | Alexander R Dunn |
| National Institutes of Health | R35GM131747 | William I Weis |
| National Science Foundation | Graduate Fellowship | Amy Wang |
| Stanford University | Stanford Graduate Fellowship | Amy Wang |
| National Institutes of Health | T32GM120007 | Amy Wang |

The funders had no role in study design, data collection, and interpretation, or the decision to submit the work for publication.

### Author contributions

Amy Wang, Conceptualization, Data curation, Software, Formal analysis, Investigation, Methodology, Writing – original draft, Writing – review and editing; Alexander R Dunn, Conceptualization, Formal analysis, Supervision, Funding acquisition, Investigation, Methodology, Writing – original draft, Project administration, Writing – review and editing; William I Weis, Conceptualization, Supervision, Funding acquisition, Investigation, Writing – original draft, Project administration, Writing – review and editing

### Author ORCIDs

Amy Wang http://orcid.org/0000-0003-4139-4563
Alexander R Dunn http://orcid.org/0000-0001-6096-4600
William I Weis http://orcid.org/0000-0002-5583-6150

### Decision letter and Author response

Decision letter https://doi.org/10.7554/eLife.80130.sa1
Author response https://doi.org/10.7554/eLife.80130.sa2

## Additional files

### Supplementary files

• Supplementary file 1. Analysis of energy-minimized models of isolated and bound actin-binding domain (ABD) superimposed with actin.

• Supplementary file 2. Kinetic parameters for a ternaryΔH1 two-state slip bond model. The estimated parameters for the purported weak state in the two-state slip bond model, $B_1$, predicted binding lifetimes an order of magnitude larger than that of ternary wild type. Akaike information criterion (AIC) and Bayesian information criterion (BIC indicate the slip-bond model better describes the data).

• Supplementary file 3. $R^2$ values of single exponential and biexponential fit to survival probability distributions in 2 pN bins. Biexponential fit $R^2$ are higher in both the ternary wild-type and ternaryΔH1 datasets, although the single exponential model describes ternaryΔH1 better ($R^2 > 0.95$) than it does for ternary wild type ($R^2 > 0.8$).

• Supplementary file 4. Two-sample Kolmogorov–Smirnov (KS) test comparing ternary wild-type versus ternaryΔH1 survival probability distributions.

• Supplementary file 5. Best-fit two-bound-state catch bond model kinetic parameters for single-step data of αE-catenin actin-binding domain (ABD) (top) and monomer (bottom).

• MDAR checklist

• Source data 1. Contacts between isolated and bound actin-binding domain (ABD) superimposed with actin.

### Data availability

All data and analysis code have been provided as zip files.

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
