## [Editor Report]

Single-molecule assays and kinetic modelling reported here validate and advance a structure-based model of the cadherin-catenin F-actin catch bond interaction, which is a fundamental cell-cell adhesive structure that can be both dynamic and force-activated. It is shown that the catch bond results from a force-dependent conformational change mechanism that may be conserved across other actin binding proteins.

---

## [Decision Letter]

**Decision letter after peer review:**

Thank you for submitting your article "Mechanism of the cadherin-catenin F-actin catch bond interaction" for consideration by *eLife*. Your article has been reviewed by 2 peer reviewers, and the evaluation has been overseen by a Reviewing Editor and Suzanne Pfeffer as the Senior Editor. The following individual involved in review of your submission has agreed to reveal their identity: Jie Yan (Reviewer #1).

Essential revisions:

Please respond to the reviewer comments and take them into account in your revised manuscript.

*Reviewer #1 (Recommendations for the authors):*

Below are my detailed comments mainly for additional discussions in a revised manuscript.

Page 5, paragraph 2, the sentence "…force on the beads commonly decreased in several discrete steps ('multi-step'), with each step corresponding to the release of a load-bearing molecule from the filament (Figure 3 —figure supplement 1)." Under tensile forces of a few pN, the domains in the α-catenin may unfold, as shown by Yao et al., (Nature communications 5, 4525, 2014). Could domain unfolding leads to similar stepwise signals? How can domain unfolding and dissociation of a complex from F-actin be distinguished?

The kinetics model shown in figure 2 contains six transitions and 12 kinetic parameters (zero-force rates and transition distances). These parameters were determined using maximum likelihood estimation from observed force-dependent binding lifetimes of a single complex (the last complex on F-actin). Considering many parameters in the model, how reliable are the determined parameters?

Page 4, the last paragraph: By energy minimization, the authors show that the actin-bound four-helix ABD structure had an increase surface contact area, supporting the proposal that the bound four-helix ABD represents a more stable bound state than the bound five-helix ABD. While this mechanism can explain the observed catch bond, alternative or additional factors need to be discussed. When the bound ABD switches from a five-helix bundle to a four-helix bundle, the stretching geometry is significantly altered. It is well known that different stretching geometries applied to the same molecule could lead to very different mechanical stability. A famous example is force dependent strand separation of a dsDNA segment – under unzipping geometry, strand separation occurs at forces in 10-20 pN at sub-second time scale, whereas under shearing geometry, strand separation occurs at forces above 60 pN at sub-second time scale. Similar stretching-geometry phenomenon may occur to protein domains, which is discussed in a recent review by Le et al., (Current Opinion in Solid State and Materials Science 25 (1), 100895, 2021).

Page 7, the sentence "H1 would be pulled away from the H2-H5 bundle more readily when force was directed towards the (-) end of F-actin but would be relatively more aligned with purported weak state when force was directed towards the barbed (+) end…" I am a little confused about the notation of the force direction. Is it the direction of the force applied to the H0 of the bound ABD, or is it the force applied to the F-actin? In figure 5-supplement figure 3, please indicate the directions of the two opposing forces, the force on F-actin and that on ABD.

It would be helpful if the authors can add a brief discussion on how the results provide an understanding of the mechanical stability of the cadherin/β-catenin/α-catenin/F-actin force-transmission linkage. At ~ 4 pN where the lifetime is the longest, the lifetime of the F-actin bound catenin complex is shorter than 10 s. In contrast, at the same level of force, the β/α-catenin connection is much more stable, which has a lifetime of > 100 seconds (Le et al., Angewandte Chemie 131, 18836, 2019). Α/β-catenin complex is also highly stable (personal communication with a peer). Does it suggest that the F-actin connection is the weakest point in the cadherin/β-catenin/α-catenin/F-actin linkage? Can the time scale enable robust mechanotransduction function? After vinculin binds and engages with the same F-actin, is the lifetime of the linkage expected to increase significantly?

*Reviewer #2 (Recommendations for the authors):*

1. In the last line of the Results section, the authors state: "… but that interactions that directly or indirectly involving b-catenin inhibit cooperative binding interactions between neighboring complexes." Do the authors really mean b-catenin here, or a-catenin? Seems that "a-catenin" better works in this context, although I recognize that b-catenin binding to a-catenin alters allostery of the latter, as shown in by this group's previous cysteine-labeling study. Please clarify.

2. Evidence that H0/H1 imposes directionality to a-cat/F-actin catch-bond mechanism is intriguing: WT a-cat favors binding to actin filaments pulled towards the minus (non-growing) rather than + (growing) end, whereas a-cat lacking H0/H1 fails to show this bias. However, given most actin structures that interface with junction structures are of mixed polarity, I struggle to understand the broader meaning of this finding. While not the job of this team to give up their working hypotheses and means of testing the broader significance of this finding, some clarity here would help.

---

## [Author Response]

Reviewer #1 (Recommendations for the authors):Below are my detailed comments mainly for additional discussions in a revised manuscript.Page 5, paragraph 2, the sentence "…force on the beads commonly decreased in several discrete steps ('multi-step'), with each step corresponding to the release of a load-bearing molecule from the filament (Figure 3 —figure supplement 1)." Under tensile forces of a few pN, the domains in the α-catenin may unfold, as shown by Yao et al., (Nature communications 5, 4525, 2014). Could domain unfolding leads to similar stepwise signals? How can domain unfolding and dissociation of a complex from F-actin be distinguished?

We thank the reviewer for prompting us to expand on this point. As the reviewer notes, it is indeed the case that the α-catenin M domain unfolds reversibly with a force midpoint of approximately 5 pN. If reversible unfolding of this (or other) domains resulted in observable transitions in our data, we would expect to see reversible jumps in force, where the optical trap beads step from, for example, 5 pN, to 4, and then back to 5 pN as the domain unfolds and then refolds. Further, these transitions should have a characteristic magnitude, which for the M domain might be ~1.4 pN at the trap stiffnesses we commonly used. Although we *do* observe such transitions, they are quite rare, to a degree that their presence is anecdotal.

Recently we performed the optical trap experiment on the ternary complex of E-cadherin, β-catenin, and α-catenin in the presence of the vinculin head domain, which binds to the extended conformation of the M domain. The single-step and last-step lifetime distributions we measured in the presence of the vinculin head domain were highly similar to those measured for the ternary complex, again indicating that our data are unlikely to reflect a large contribution from reversible unfolding events (Bax et al. 2022, Figure 2).

Extending this logic, it is likely that the multi-step events correspond to sequential rupture of several complexes from actin rather than domain unfolding. In support of this, Buckley et al., 2014 observed an increase in single-step events after adding soluble ABD, a result that would be expected if the experiment were probing detachment events, but not if the observed force transitions reflected a conformational transition in the cadherin-catenin complex.

The kinetics model shown in figure 2 contains six transitions and 12 kinetic parameters (zero-force rates and transition distances). These parameters were determined using maximum likelihood estimation from observed force-dependent binding lifetimes of a single complex (the last complex on F-actin). Considering many parameters in the model, how reliable are the determined parameters?

We have done our best to assess the uncertainties associated with the individual parameters (Table 1, 2, S2, S5). Before discussing this, we should note a small clarification motivated by this feedback. The fit model contains 8 free parameters, as on rates are not included. We’ve modified Figure 2, Figure 5, and Figure 5 – supplement 3 to make this more readily apparent. We also updated the methods section to include that we constrained our 8-parameter fit to the two-state catch bond model such that the mean lifetime at zero force is less than or equal to 100 s.

Assessing fit uncertainties with an 8-parameter model is not trivial. To adequately account for parameter co-variation, we used a bootstrap resampling approach (model fitting and confidence intervals (CIs) section). Parameters fall in physical reasonable ranges, though some like k2→10 and x2→1 are poorly constrained. Physically, this appears to reflect a tendency of the model to restrain the equilibrium constant between the weak and strong states, rather than the individual rates.

Despite its complexity, the two-state catch bond model is necessary to capture the biphasic lifetime distributions we observe (Figure 5 —figure supplement 2; also Buckley et al., 2014) for the wildtype complexes, as well as the asymmetry in lifetimes with respect to the direction of applied load. We likewise observe agreement between k2→0 for the ternary wild-type and the dissociation rate for ternaryΔH1 (Table 2, highlighted rows), suggesting that both measure dissociation from a similar strongly bound state.

Page 4, the last paragraph: By energy minimization, the authors show that the actin-bound four-helix ABD structure had an increase surface contact area, supporting the proposal that the bound four-helix ABD represents a more stable bound state than the bound five-helix ABD. While this mechanism can explain the observed catch bond, alternative or additional factors need to be discussed. When the bound ABD switches from a five-helix bundle to a four-helix bundle, the stretching geometry is significantly altered. It is well known that different stretching geometries applied to the same molecule could lead to very different mechanical stability. A famous example is force dependent strand separation of a dsDNA segment – under unzipping geometry, strand separation occurs at forces in 10-20 pN at sub-second time scale, whereas under shearing geometry, strand separation occurs at forces above 60 pN at sub-second time scale. Similar stretching-geometry phenomenon may occur to protein domains, which is discussed in a recent review by Le et al., (Current Opinion in Solid State and Materials Science 25 (1), 100895, 2021).

We agree with the reviewer with regard to the general principle. In this case, we feel that the effect of pulling geometry is most clearly demonstrated by the effect of deleting of H0 and H1 on catch bond directionality. We address this point in the Discussion as follows:

“These data indicate an additional role for H0 and H1 in modulating the relative orientation of applied force and the reaction coordinate that characterizes the transition between the weak and strong states. More broadly, the rearrangement of the ABD from a five to four helix bundle necessarily alters the axis along which force is applied. This might be expected to alter the force-dependent actin dissociation of the strong versus weak state (Le et al., 2021).”

Page 7, the sentence "H1 would be pulled away from the H2-H5 bundle more readily when force was directed towards the (-) end of F-actin but would be relatively more aligned with purported weak state when force was directed towards the barbed (+) end…" I am a little confused about the notation of the force direction. Is it the direction of the force applied to the H0 of the bound ABD, or is it the force applied to the F-actin? In figure 5-supplement figure 3, please indicate the directions of the two opposing forces, the force on F-actin and that on ABD.

We modified Figure 5 —figure supplement 3 to demonstrate that the restoring force on the actin filament is transduced to the bound ABD.

It would be helpful if the authors can add a brief discussion on how the results provide an understanding of the mechanical stability of the cadherin/β-catenin/α-catenin/F-actin force-transmission linkage. At ~ 4 pN where the lifetime is the longest, the lifetime of the F-actin bound catenin complex is shorter than 10 s. In contrast, at the same level of force, the β/α-catenin connection is much more stable, which has a lifetime of > 100 seconds (Le et al., Angewandte Chemie 131, 18836, 2019). Α/β-catenin complex is also highly stable (personal communication with a peer). Does it suggest that the F-actin connection is the weakest point in the cadherin/β-catenin/α-catenin/F-actin linkage? Can the time scale enable robust mechanotransduction function? After vinculin binds and engages with the same F-actin, is the lifetime of the linkage expected to increase significantly?

The connection to F-actin is almost certainly the weakest link. Our solution (i.e. no force) measurements show that the dissociation constant of the ABD and F-actin is approximately 10 μM, and the H0/H1 deletion lowers this dissociation constant 200 nM. The presence of β-catenin further weakens this interaction for the full-length α-catenin. Therefore, even if we take 200 nM as the strongest affinity (i.e. either the peak lifetime at 6 pN force or the H0/H1 deletion at 0 force shown here), the affinity is still roughly an order of magnitude weaker than the α-catenin/β-catenin interaction. Prior measurements likewise show that α-catenin binds to the β-catenin/cadherin complex with an affinity of about 1 nM, and β-catenin binds to cadherin with affinities ranging from 50 pM to 40 nM depending on the phosphorylation state of cadherin (Choi et al., 2006; Pokutta et al., 2014).

The timescale issue is an interesting one that, to our knowledge, has not been probed experimentally. We note that the cadherin-catenin binding lifetime under force increases roughly 10-fold under load, and that cooperative interactions between neighboring cadherin-catenin complexes lead to an additional increase in binding lifetimes (Bax et al., 2022). Finally, as the reviewer notes, M domain unfolding and subsequent recruitment of vinculin further stabilizes junctions, a finding that is supported by cell biological measurements (Yonemura et al., 2010, Han et al., 2016). Thus, from an equilibrium standpoint, it does seem that properties of the system are well evolved to mediate mechanotransduction.

We have added the following to the Discussion section, paragraph 6 to clarify this point:

“Nonetheless, tension-dependent changes in N and M domains associated with αE-catenin binding to actin on the timescale of seconds may enable recruitment of actin-binding partners (Yao et al., 2014; Yonemura et al., 2010), such as vinculin, which can additionally mediate dynamic linkages and organization at cell junctions (Bax et al., 2022).”

Reviewer #2 (Recommendations for the authors):1. In the last line of the Results section, the authors state: "… but that interactions that directly or indirectly involving b-catenin inhibit cooperative binding interactions between neighboring complexes." Do the authors really mean b-catenin here, or a-catenin? Seems that "a-catenin" better works in this context, although I recognize that b-catenin binding to a-catenin alters allostery of the latter, as shown in by this group's previous cysteine-labeling study. Please clarify.

We revised the text to clarify this point accordingly:

“[…] but that interactions involving the N and M domains of αE-catenin, as well as with β-catenin, inhibit cooperative binding interactions between neighboring complexes”.

2. Evidence that H0/H1 imposes directionality to a-cat/F-actin catch-bond mechanism is intriguing: WT a-cat favors binding to actin filaments pulled towards the minus (non-growing) rather than + (growing) end, whereas a-cat lacking H0/H1 fails to show this bias. However, given most actin structures that interface with junction structures are of mixed polarity, I struggle to understand the broader meaning of this finding. While not the job of this team to give up their working hypotheses and means of testing the broader significance of this finding, some clarity here would help.

We thank the Reviewer for the opportunity to clarify this important point.

While mixed actin polarity is observed at junctional structures, the actin cables in some epithelial tissues show clear sarcomeric organization, implying that the barbed, (+)-ends of the filaments terminate at tricellular junctions (Ebrahim et al., 2013, Yu-Kemp et al., 2022, Coravos et al., 2016, Houssin et al., 2020, Gomez et al., 2015). This arrangement is consistent with cell biological, genetic, and electron microscopy data indicating that actin filaments are anchored end-on at epithelial tricellular junctions (Choi et al., 2016, Yonemura et al., 2011). A closely related cadherin-catenin complex likewise links myofibrils across the junctions between cardiomyocytes in the heart (e. g. Li et al., 2019). In each of these cases, F-actin (+)-ends must be selectively anchored at the cadherin-catenin complex for efficient force transduction to occur. It is intriguing to speculate that the asymmetric binding interaction of α-catenin with F-actin, which favors (+)-end binding, may help to seed and/or reinforce force-generating sarcomeric order. More broadly, we have recently found that vinculin (Huang et al., 2017) and talin (Owen et al., 2022) also form directionally asymmetric catch bonds to F-actin. Because the ABDs of these proteins are structurally similar to that of αE-catenin, a conserved mechanism for directional catch bond behavior may be a common characteristic of both integrin- and cadherin-based adhesion complexes.

The above points are treated in detail in Bax et al., which is available on bioRxiv and is currently in revision. Because it is not the main point of this manuscript, we address the issue of directionality more in relation to similar observations for vinculin and talin:

We now address this point briefly in the Discussion:

“Although the biological function of directional catch bond formation remains speculative, its presence across multiple adhesion proteins is striking. One possibility is that the force-dependent directionality imparted by these interactions may serve to initiate or reinforce long-range order in the actin cytoskeleton, a possibility that is consistent with cell biological data and theoretical modeling (Bax et al., 2022; Huang et al., 2017; Rahman et al., 2016).”